# Effects of Botulinum Toxin on Migraine Attack Features in Chronic Migraine: A Six-Month Open-Label Observation Study through Electronic Diary Smartphone Application

**DOI:** 10.3390/toxins11110668

**Published:** 2019-11-15

**Authors:** Antonio Santoro, Marianna Delussi, Maurizio Leone, Anna Maria Miscio, Laura De Rocco, Gianluca Leo, Marina De Tommaso

**Affiliations:** 1Casa Sollievo della Sofferenza San Giovanni Rotondo Foggia, 70013 San Giovanni Rotondo, Italy; antonio.santoro@operapadrepio.it (A.S.); m.leone@operapadrepio.it (M.L.); anna.miscio@gmail.com (A.M.M.); 2Applied Neurophysiology and Pain Unit, SMBNOS Department, Bari Aldo Moro University, 70121 Apulia, Italy; marina.detommaso@uniba.it; 3Terin Consortium, Mesagne, 72100 Brindisi, Italy; laura.derocco@terin.it (L.D.R.); gianluca.leo@thcs.it (G.L.)

**Keywords:** chronic migraine, botulin toxin, electronic headache diary, headache smartphone application

## Abstract

OnobotulintoxinA (OBT-A) is a treatment option for Chronic Migraine (CM). It works on central sensitization and pain but its mode of action is still unknown. To observe how OBT-A treatment works on single migraine attacks, this paper covers an over-6-month observation period through self-reported smartphone application data. This was an observational, open-label cohort study conducted on 34 CM patients under OBT-A treatment, selected between December 2016 and December 2017, who agreed to download a smartphone headache diary application (Aid Diary) according to the study instructions. The analysis was conducted using the smartphone application data reports on allodynia, intensity and extension of pain, and vegetative symptoms. We analysed a total of 707 records of single migraine attacks reported by compliant users (n = 34) in real-time. OBT-A significantly reduced allodynia, the number of vegetative symptoms, pain extension and intensity in single migraine attacks. Pain intensity was correlated with pain extension. In single migraine attacks, OBT-A improved symptoms of central sensitization. This action could be exerted by modulating nociceptive transmission and reducing the burden of single migraine episodes and improving the overall quality of life.

## 1. Introduction

Chronic Migraine (CM) is an invalidating disease caused by frequently recurrent migraine attacks and pain persistence between migraine days. Central sensitization facilitates pain persistence, leading to functional changes within the nociceptive trigeminal caudal and cervical system [1]. The efficacy of Onabotulinumtoxin A (OBT-A) for the prophylaxis of CM has been demonstrated over recent years in the well-designed Phase III Research Evaluating Migraine Prophylaxis Therapy (PREEMPT) 1 and 2 trials [2,3]. However, the mechanism of action of OBT-A and the optimal clinical characteristics of responders are not yet clear. Studies on animal models and healthy subjects supported the hypothesis that botulinum toxin could act on nociceptive afferents, modulating central transmission and reducing the development of central sensitization [4,5,6]. Our group recently demonstrated that OBT-A might modulate the lack of habituation of trigeminal nociceptive responses in chronic migraine patients [7]. Reduced habituation to repetitive painful stimuli may be the counterpart of central sensitization at the trigeminal level. This neurophysiological effect was correlated with clinical efficacy [7].

In a more recent study, we evaluated the long-term effect of OBT-A in CM and observed a reduction of allodynia and global burden of migraine [8].

The knowledge of the clinical features of migraine attacks is mainly based on retrospective studies. In the current scientific literature, no study analyzed the short-term effects of OBT-A single migraine attack features through mobile phone application reports. The mobile phone app could be a useful tool to extemporarily record migraine features. In today’s digital environment, individuals are increasingly using apps to regularly record pain, nutrition, and various medical information [9].

Here we present an open-label six-months-observational study, based on reports of single migraine attacks from real-world CM patients using a self-reported smartphone application, the Aid Diary. Specific aims of the study were the live detection of single migraine attack characteristics, such as pain intensity, allodynia, and pain extension, just to better understand the mode of action of OBT-A on central sensitization symptoms and single migraine attack features.

## 2. Results

Sample Description: The study sample included 707 migraine records (i.e., migraine attacks) from 34 users who satisfied the inclusion criteria for the study, were compliant with given instructions and received two OBT-A treatments during six months.

All included records satisfied the features of migraine without aura [10,11].

A summary of demographics and migraine characteristics of the study sample is shown in Table 1 and Figure 1.

The number of headache episodes, resulting from the paper diary, progressively decreased at T1 and T2 compared to T0 (T0 19.88 ± 6.28; T1 11.07 ± 4.48; T2 6.63 ± 2.42. ANOVA F 56.23 *p* < 0.0001; Bonferroni test T0 vs. T1 *p* < 0.0001; T0 vs. T2 *p* <0.0001; T1 vs. T2 *p* < 0.012). No patient reported relevant side effects. Two patients complained about slight neck pain after the first cycle of OBT-A injections.

The number of allodynia symptoms, as described by the application, significantly decreased for effect of condition and the post-hoc Bonferroni test was significant for T0 vs. T1. The topographic distribution of pain was limited to significantly fewer points at T1 and T2 compared to T0. A reduction of the vegetative symptoms was observed at T1 with respect to T0. The intensity of pain significantly decreased after the first and the second round of injections. All outcome measures are shown in Table 2.

## 3. Discussion

The present study evaluated for the first time the features of single migraine attacks in patients treated with OBT-A, who used an innovative method to report the severity of acute symptoms in real-time [12].

In previous studies on CM patients, OBT-A treatment reduced the number of days of headache and migraine, the consumption of triptans and disability, improving the overall quality of life of migraine patients [13].

On top of the OBT-A known effects on headache frequency and disability in CM patients, our results highlight how OBT-A treatment works on clinical features of single migraine attacks expressed by pain intensity, allodynia, vegetative symptoms, and pain extension. The digital app allowed the analysis of data that were most likely to be truthful, since migraine characteristics were recorded in real-time, thus avoiding the bias affecting questionnaires or paper diaries [9]. The real-time reports of patients confirmed that OBT-A can improve the severity of single migraine episodes, reducing the impact of central sensitization symptoms, just after the first session of treatment. Allodynia symptoms were lower than baseline also after the second session, though the statistical difference was not significant, probably for the reducing effect of OBT-A on the total number of migraine episodes.

The smartphone app also allowed, for the first time, evaluation of the global extension of pain in the cranial and pericranial sites in single migraine attacks. Patients perceived a decrease of the painful area after the first and second OBT-A injections rounds. This effect could be also attributable to a reduction of central sensitization phenomenon at the second-order wide range neurons levels in the caudal and cervical nuclei, with an improvement of the scalp and neck areas hyperalgesia [6].

The total number of vegetative symptoms seemed generally reduced in single migraine attacks, though this effect was not significant in the comparison between the T2 phase and baseline condition, probably due to the fewer migraine attacks after the second round of OBT-A injections, as above reported. No study described in detail the outcome of vegetative symptoms after OBT-A injections in chronic migraine patients. Our patients sent their reports before the intake of symptomatic treatment. The vegetative involvement was consistent and compatible with the migraine diagnosis. OBT-A reducing effect on vegetative symptoms may result from its modulating action on trigeminal function, with a reduced gain of the trigeminovascular system as a complex, and subsequent partial inhibition of sphenopalatine ganglion activation [14].

According to the theory of the extracranial representation of C fibers converging on the caudal trigeminal nuclei, the OBT-A injection could cause a trigeminal nociceptors inhibition at peripheral level [14]. This inhibition mechanism could lead to a progressive change of trigeminal function, with a decreased level of activation and consequent reduction of peripheral and central sensitization, as well as activation of vegetative reflex responses. According to this hypothesis, subjective pain intensity decreased after OBT-A injections due to the possible reduction of gain of trigeminal activation. The subjective perception of pain decreased also for the restriction of hyperalgesia extension, as emerged from the positive correlation between VAS values and the number of topographic coordinates. We previously observed the effect of OBT-A on the neurophysiological signature of central sensitization phenomena, expressed by habituation of pain-related cortical responses to trigeminal stimulation [7] and OBT-A long-term effect on central sensitization, expressed by allodynia, with reduction of the global burden of disease [8]. The present results provided clinical evidence of an attenuation of the strength of trigeminal activation carried out by OBT-A, with a consequent reduction of the central mechanism of pain diffusion, pain intensity, and vegetative nuclei activation.

### Study Limitations

Studies using self-reported data from a digital app have some known inherent limitations. The first is the poor compliance of patients throughout the monitoring period; 23 out of 57 (43.35%) enrolled patients dropped out during a monitoring phase sometimes due to technical reasons related to the updates required to have the application properly working. Secondly, patient subjective judgment about the type of headache to be reported—e.g., migraine or tension-type—could be mistaken. However, the diagnosis of chronic migraine is based on the subjective statement of patients about the number of migraine or tension-type episodes [10,11] The description of acute symptoms satisfied the criteria for migraine attacks in any case. The simple description of subjective symptoms could suggest the effect of OBT-A on trigeminal activation in single migraine attacks, while the direct observation of patients in the acute state could give the definite confirmation of present results. The study followed an open real-life design, but the absence of a placebo control could limit the reliability of the reported migraine attack characteristics. The placebo effect is relevant after toxins injections [15], but the most recent meta-analysis confirmed the superiority of OBT-A in the prevention of migraine attacks occurrence and severity [16]. Additional important migraine features, such as the response to symptomatic treatment or the quality of migraine, i.e., imploding rather than exploding headache [14], could improve the reliability of this smartphone application.

## 4. Conclusions

In summary, the present study provided evidence for the first time about the change of single migraine attack features after OBT-A injections. Taking into account the above-described limitations, the OBT-A effect on migraine intensity, pain diffusion, and vegetative symptoms confirms a global modulatory action on the strength of trigeminal activation. Another potential value of the study could be the reliability of the smartphone app, which deserves further upgrades for the final confirmation of its value in the clinical assessment of migraine. These applications would be potentially useful for patients, their caregivers, or physicians to improve the management of the disease, but also for the wider scientific community to increase the knowledge about the way of action of treatments on acute migraine features.

## 5. Materials and Methods

### 5.1. Patients

This was an observational, open-label, cohort study conducted in accordance with the principles of the Helsinki Declaration. The study included 57 CM patients that spontaneously came to San Giovanni Rotondo Casa Sollievo della Sofferenza Hospital (San Giovanni Rotondo, Foggia, Italy) between December 2016 and December 2017, agreed to OBT-A treatment (Botox-producing laboratory Wespole Ireland, registered in Italy) and to be enrolled in our study conducted through a smartphone headache diary application. The Aid Diary was specifically created to capture the extempore details regarding clinical features of migraine by a consequential series of questionnaires.

Neurologists with experience in headache management selected CM patients for OBT-A treatment. CM patients were diagnosed according to the most recent classifications [10,11]. The medical staff reviewed the diagnosis also retrospectively, considering the most recent criteria [11]. Exclusion criteria were non-compliance with OBT-A treatment in the six-month observation period, psychiatric diagnosis, according to DSMV, other neurological diseases, including other forms of primary headaches and Medication Overuse Headache [11], and systemic diseases.

In line with the Italian Public Health System rules, patients were treated with OBT-A after at least one failed preventive treatment, among those reported by the Italian guidelines [17]. In some patients with a slight therapeutic effect, preventative drugs were continued during OBT-A treatment.

All patients received a paper diary and a smartphone application headache diary at the time of visit booking, before the first session of OBT-A treatment. Users were instructed to recognize migraine characteristics, reporting all headache attacks on their paper diary and only migraine attacks on smartphone application before taking symptomatic therapy with triptans or pain-reducing drugs. Patients were also instructed to fill out both headache diaries 30 days before the day of first OBT-A treatment; moreover, they were instructed to follow the directions of the app minutely, not to skip any questionnaires nor to miss any information.

We included in the present study the patients who received at least two OBT-A treatment cycles during the six months. Moreover, we included into the data analysis only compliant patients, who completed the electronic diaries reporting the days of migraine attacks with relative features and symptoms.

All patients signed informed consent for the inclusion of their clinical and demographic data in the observational study, with respect of their privacy and the treatment of personal anonymized data. The study was approved by the local Ethic Committee of Bari-Policlinico General Hospital. The approval code: 7/3423.

Considering the observational nature of the study, we did not preliminarily calculate the sample size.

### 5.2. The Smartphone Application

A smartphone app named Aid Diary was specifically designed for this study, by Terin Consortium (http://www.terin.it/) It was built to be easy to use, with an obligatory sequence of questionnaires to tick in order to send a complete report; the questionnaires presented in order are: Visual Analogue Scale (VAS) for pain (Figure 2), Allodynia Questionnaire [8,13,18,19], a list of vegetative symptoms associated with migraine attack (nausea, vomiting, phono-photophobia) [11,12], and an image of the head, neck and other parts of the body, in frontal and posterior view, where the patient had the possibility to click exactly on the sites where he/she perceived the pain, to render a report of the extension of headache according to precise and predetermined coordinates (Figure 3). For allodynia, 12 total symptoms were reported, in accordance with previous studies [8,13,18,19], so the final score between 0 and 12 indicated the number of acute allodynia features. For vegetative symptoms, three items were considered, with a final score between 0 and 3. For pain extension, the body and the head figure were divided in points, so the final score expressed the number of points the patients indicated by touching the screen in correspondence to the painful region. The higher the number of points, the more extended the painful area. The VAS values were expressed in a numerical scale between 0 (no pain) and 10 (maximum pain). Each migraine record describes a single migraine attack with all the above characteristics.

We monitored the progress of the reports of each single user through a virtual platform (http://ihcs.terin.it/home) that allowed us to view all the reports and related items in real-time. This virtual platform shows the individual reports and graphs of any questionnaires (Figure 3).

Data from users who sent records by Aid Diary App for less than two consecutive weeks or recorded less than 15 migraines during the observation period or skipped some questionnaires were excluded.

### 5.3. OBT-A Treatment

One hundred and fifty-five/195 U of OBT-A were injected every three months, into 31 to 39 pericranial sites, with additional injections in the sites of predominant pain location in up to three specific muscle areas (i.e., occipitalis, temporalis, and trapezius). [2,3]. We used 155 U in the first treatment cycle, and 195 U with additional sites in the following cycle in all cases.

For the present analysis, we considered data collected at baseline T0 (30 days before first injection session), T1 (records in the 90 days following first OBT-A treatment), and T2 (records in the 90 days following second OBT-A treatment session).

## 6. Statistical Analysis

For the aim of this study, we entered the data from single migraine attacks into an SPSS database, as variables of univariate ANOVA to be compared among the T0, T1 and T2 conditions. The post-hoc Bonferroni Test was computed.

The variables for ANOVA analysis were single attacks’ number of allodynia symptoms, the topographic distribution of pain with the total number of coordinates, number of vegetative symptoms, and subjective pain intensity as measured by a 0–10 numerical scale. (Table 2)

The frequency of headache, detected by paper diaries, was compared among T0, T1, and T2 conditions, using the repetitive measure ANOVA and single contrasts computation.

The Pearson correlation test was used to measure the correlation between the different migraine attack features.

For statistical analysis, we used the SPSS IBM vers.21 (Armonk, NY, USA).

## Figures and Tables

**Figure 1 toxins-11-00668-f001:**
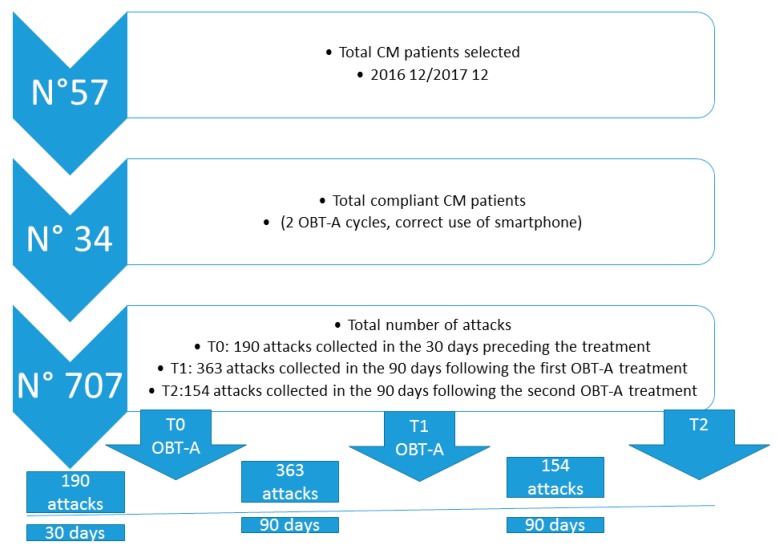
Flow chart reporting the total numbers of patients treated with OBT-A, compliant subjects, attacks eventually analyzed, and number of OBT-A cycles requested for study inclusion.

**Figure 2 toxins-11-00668-f002:**
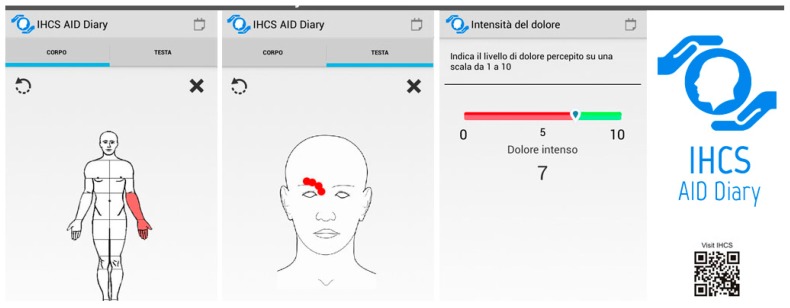
Pain extension on the head and the body and Visual Analogue Scale (VAS) as represented in the Aid application. The original instructions in Italian language are reported.

**Figure 3 toxins-11-00668-f003:**
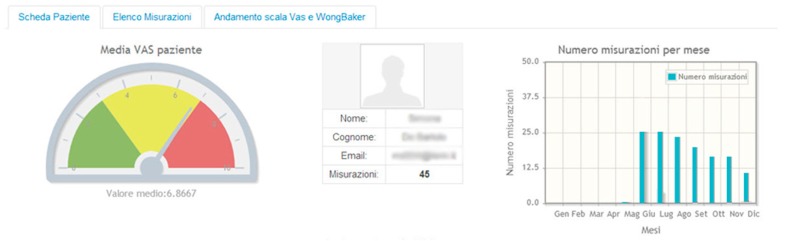
The virtual platform (http://ihcs.terin.it/home) shows the summary of all reports and mean VAS values in a representative CM patient. The original legends in Italian language are reported.

**Table 1 toxins-11-00668-t001:** Demographics and Migraine Information of users included in the study. CM: chronic migraine Users.

Characteristic	CM (n = 34)
Female (n) *	29
Male (n) *	5
Age Mean ± SD *	41.8 ± 9.4
Migraine records (total)	707
Migraine records in T0 (30 days preceding first OBT-A treatment)	190
Migraine records in T1 (90 days following first OBT-A treatment)	363
Migraine records in T2 (90 days following second OBT-A treatment)	154
Mean migraine records/patient **	207

* Numbers of CM Users. ** Total number of migraine records divided by total number of CM Users.

**Table 2 toxins-11-00668-t002:** Statistical analysis by univariate (ANOVA) and outcome measures of single attack migraine features from all CM patient Users (n = 34).

ALLODYNIA	T0	T1	T2
Mean	3.082	2.416	2.53
Standard Error	0.188	0.116	0.193
ANOVA	F 4.552 df 2 *p* < 0.011		
Bonferroni test T0 vs. T1 T0 vs. T2	*p* < 0.008 *p* < 0.134		
VEGETATIVE SYMPTOMS	T0	T1	T2
Mean	2.817	2.393	2.278
Standard Error	0.124	0.083	0.134
ANOVA	F 4.939 df 2 *p* < 0.008		
Bonferroni T0 vs. T1 T0 vs. T2	*p* < 0.014 *p* < 1		
PAIN EXTENSION	T0	T1	T2
Mean	10.67	6.5	3.93
Standard Error	9.25	8.214	3.507
ANOVA	F 7.597 df 2 *p* < 0.001		
Bonferroni test T0 vs. T1 T0 vs. T2	*p* < 0.014 *p* < 0.001		
VAS Pain	T0	T1	T2
Mean	6.84	5.29	5.75
Standard Error	2.106	3.587	2.898
ANOVA	F 15.565 df 2 *p* < 0.0001		
Bonferroni test T0 vs. T1 (p) T0 vs. T2 (p)	*p* < 0.0001 *p* < 0.004		

VAS pain values were significantly correlated with pain extension in the total of reports (Pearson Correlation Test: VAS pain vs. pain extension 0.302, *p* < 0.0001).

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
