# Peer review of "Effects of Botulinum Toxin on Migraine Attack Features in Chronic Migraine: A Six-Month Open-Label Observation Study through Electronic Diary Smartphone Application"

_toxins, 2019, doi:10.3390/toxins11110668_

Round 1
Reviewer 1 Report
The authors addressed all my queries and the manuscript in the present form can be published.
Author Response
Thank you for your comments
We further corrected typos and further improve English style
Reviewer 2 Report
the changes have been made successfully
However, Figure 2 should be removed.the changes have been made successfully
Author Response
Thank you for the consideration of our work
We removed Figure 2, according to your suggestion
This manuscript is a resubmission of an earlier submission. The following is a list of the peer review reports and author responses from that submission.
Round 1
Reviewer 1 Report
The present study evaluated the features of single migraine attacks in patients under OBT-A treatment, who used an innovative method to report in real time the severity of acute symptoms Another prospective value of the study could be the reliability of the smartphone app, potential utility for patients, their caregivers or physicians to better manage the disease
The topic of the study of this paper is very interesting .This was an observational, open-label and the absence of placebo control could limit the reliability of the reported migraine attacks characteristics.
The trade name of the product should be added OBT-A
Table 2 is poor
typographical errors(The Person correlation ??
References bibliographies are not uniformly cited
Call attention that the section of material and methods is almost at the end of the article, appears in the journal guide
Author Response
Thank you for your comments. The requested changes are outlined in yellow color
We added the trade name of the product in the introduction.
"Table 2 is poor"
We modified the format of table 2
"Typographical errors(The Person correlation ??"
We corrected typographical errors
"References bibliographies are not uniformly cited"
We adjusted the reference format
"Call attention that the section of material and methods is almost at the end of the article, appears in the journal guide"
We put materials and methods at the end of the manuscript
Reviewer 2 Report
Review on manuscript:
Effects of Botulinum Toxin on migraine attack features in Chronic Migraine: a six-months open label observation study through electronic diary smartphone application.
The authors report that the treatment with onobotulilnum toxin A (OBT-A) effectively reduces migraine sensitization. This is a six-months open labelled study with observational design. However, the authors gave their aim of the study: they tried to find out how OBT-A exerts this effect. Unfortunately, at least from my point as reviewer, it is very difficult or even not possible to deduce firm informations on mechanisms of drugs actions from an observational study with 34 participants. The authors discuss their results correctly from this point of view but there is no real evidence – it’s (although probable) only a speculation. This instance and more detailed information on limitations (additionally to the already correctly discussed limitations) of the study should be very clearly stated.
Additionally, the statement “The study was descriptive in nature and thus no formal hypothesis testing was used for sample size calculation” is puzzling. In science (with mechanistical approach) there is always a H0 and H1 and has nothing to do with sample size calculations. I strongly recommend to reconsider this important point. But I do agree, that the calculation of a sample size would be very difficult for such an observational study.
Major comments:
- The results are surprisingly clear, at least from the statistical point of view and the number of participants included within the study. Please describe exactly how the statistics have been calculated. Especially, have all the record data put into the statistics ore means of records of each patient?
- the authors state that the study was in accordance with the principle of Helsinki Declaration and that participants signed an informed consent. There is an approval by the local Ethics Committee – nevertheless, is there more details on the information of the patients their rights etc?
- a clear figure of the patient flow (numbers of patients that have been screened for the study, who was included, how many gave their consent, how many rejected and dropouts etc.) is missing
- it’s bit surprising to me that there are no informations about “Conflicts of interest” – even in anonymous way (e.g. summarized) this should be clearly stated
- are there any safety data? Even in an observational study these data should be collected and some information about this is important.
- there is consense that placebo effects are strongly involved in the therapy of pain disorders. Please take this circumstance into account in the discussion
Minor comments:
- please provide detailed information on abbreviations
e.g. in Table 2 (what exactly is “error DS”? – I can imagine what the authors mean but it has to be specified)
- please provide a Figure with the results of the study (for the purpose of better comprehension)
Against this background it was very difficult to give a recommendation to the editor. Nevertheless, the study results are interesting and important, this my recommendation is MAJOR REVISION.
Author Response
"The authors report that the treatment with onobotulilnum toxin A (OBT-A) effectively reduces migraine sensitization. This is a six-months open labelled study with observational design. However, the authors gave their aim of the study: they tried to find out how OBT-A exerts this effect. Unfortunately, at least from my point as reviewer, it is very difficult or even not possible to deduce firm informations on mechanisms of drugs actions from an observational study with 34 participants. The authors discuss their results correctly from this point of view but there is no real evidence – it’s (although probable) only a speculation. This instance and more detailed information on limitations (additionally to the already correctly discussed limitations) of the study should be very clearly stated."
We thank the reviewer for his interesting and useful comments. We reinforced the concept of the present limitations of the study in the dedicated section. All the requested changes were outlined in green color.
"Additionally, the statement “The study was descriptive in nature and thus no formal hypothesis testing was used for sample size calculation” is puzzling. In science (with mechanistical approach) there is always a H0 and H1 and has nothing to do with sample size calculations. I strongly recommend to reconsider this important point. But I do agree, that the calculation of a sample size would be very difficult for such an observational study."
We modified the sentence, in accord with the difficulty in define sample size in observational studies.
Major comments:
-" The results are surprisingly clear, at least from the statistical point of view and the number of participants included within the study. Please describe exactly how the statistics have been calculated. Especially, have all the record data put into the statistics ore means of records of each patient?"
Single data from each migraine attack were introduced in the statistical analysis, as we better defined in the revised method section.
-" the authors state that the study was in accordance with the principle of Helsinki Declaration and that participants signed an informed consent. There is an approval by the local Ethics Committee – nevertheless, is there more details on the information of the patients their rights etc?"
We provided editor for original consensus signed by patients, as regard to their rights in terms of privacy and personal data treatment. We better specified this point in the revised method section
- "a clear figure of the patient flow (numbers of patients that have been screened for the study, who was included, how many gave their consent, how many rejected and dropouts etc.) is missing"
We provided for the requested figure.
"- it’s bit surprising to me that there are no informations about “Conflicts of interest” – even in anonymous way (e.g. summarized) this should be clearly stated"
We completed the conflict of interest section
"- are there any safety data? Even in an observational study these data should be collected and some information about this is important."
We added further information about this point .
"- there is consense that placebo effects are strongly involved in the therapy of pain disorders. Please take this circumstance into account in the discussion."
In the discussion , the lack of placebo control was present in the original version of the manuscript. Now we added further comments and references about
Minor comments:
"- please provide detailed information on abbreviations
e.g. in Table 2 (what exactly is “error DS”? – I can imagine what the authors mean but it has to be specified)"
We modified Table 2, in accord to reviewer 1, including Standard Errors in place of Error DS
"- please provide a Figure with the results of the study (for the purpose of better comprehension)"
We provided for the suggested figure
Reviewer 3 Report
The paper is written in such a poor English that it is impossible to review in the present form.
Author Response
"The paper is written in such a poor English that it is impossible to review in the present form"
We apologies for the poor English style and submitted the manuscript again to an English lecturer of our University, in order to further correct the style errors.
Round 2
Reviewer 1 Report
This was an observational, open-label, cohort study conducted on 34 CM patients in OBT-A treatment. This study evaluated for the first time the features of single migraine attacks in patients treated with OBT-A (Botox), who used an innovative method to report in real time the severity of acute symptoms
There is no conclusion in the abstract
OBT-A (Botox), I should add the name and laboratory, citizen registration ….
Fig 2 is not labeled, missing values, incomplete
Author Response
Dear Reviewer,
thank you for your comments
we changed the abstract, according to your suggestion (it is included in the revised manuscript)
we completed figure 2 with the results of Bonferroni test.
we also completed the details of Botox drug.
Reviewer 3 Report
I am really sorry, but the manuscript language is still need to be improved. It may the English correct for a non scientist, but the scientific meaning still not understandable. Please let the manuscript to correct with a professional native speaker.
Author Response
Thank you for your comments
We submitted the manuscript again to a professional English editing, hoping to have improved it as you requested
Round 3
Reviewer 3 Report
The present paper analyses the effect of OnobotulintoxinA (OBT-A) on chronic migraine. The analysis was performed in three different timepoints, T0, T1 and T2. The authors used a smartphone application (Aid Diary) for the analysis. The aim of the study is interesting and the way of collecting and analyzing the data is very modern.
However, I have several concerns about the data presentation. It is not clear how the patients were treated. The Fig1, which shows the treatment and date collecting strategy is absolutely unclear. Furthermore, it has to be clarified when the attacks (presented on the figs and in the table1) happened. It is recorded during the treatment or before the treatment? If it happened before the treatment, what does the T0, T1 and T2 means? What exactly the measured symptoms mean and how they were analyzed? Is it a grade or the duration of allodynia? How the vegetative symptoms were detected? The presented values are scores or mean values? How they were summarized? What means here T0, T1 and T2? The same questions go to the “pain extension” and “VAS pain”.
These questions have to be clarified to understand the discussion and before the manuscript can be published.